# The Law and NLP: Bridging Disciplinary Disconnects

**Robert Mahari**
MIT and Harvard Law School
rmahari@mit.edu

**Dominik Stammbach**
ETH Zurich
dominsta@ethz.ch

**Elliott Ash**
ETH Zurich
ashe@ethz.ch

**Alex 'Sandy' Pentland**
MIT
pentland@mit.edu

## Abstract

Legal practice is intrinsically rooted in the fabric of language, yet legal practitioners and scholars have been slow to adopt tools from natural language processing (NLP). At the same time, the legal system is experiencing an access to justice crisis, which could be partially alleviated with NLP. In this position paper, we argue that the slow uptake of NLP in legal practice is exacerbated by a disconnect between the needs of the legal community and the focus of NLP researchers. In a review of recent trends in the legal NLP literature, we find limited overlap between the legal NLP community and legal academia. Our interpretation is that some of the most popular legal NLP tasks fail to address the needs of legal practitioners. We discuss examples of legal NLP tasks that promise to bridge disciplinary disconnects and highlight interesting areas for legal NLP research that remain underexplored.

## 1 Introduction

Rapid advances in NLP technology are already promising to transform society and the economy (see e.g., OpenAI, 2023; Eloundou et al., 2023; Bommasani et al., 2021), not least by their impact on many professions. Given that legal practice is embedded in written language, it stands to gain immensely from the application of NLP techniques. This promise has attracted research on NLP applications related to a wide array of legal tasks including legal research (Huang et al., 2021; Ostendorff et al., 2021), legal reasoning (Guha et al., 2023; Mahari, 2021), contract review (Hendrycks et al., 2021; Leivaditi et al., 2020), statutory interpretation (Nyarko and Sanga, 2022; Savelka et al., 2019), document review (Yang et al., 2022; Zou and Kanoulas, 2020), and legal question answering (Vold and Conrad, 2021; Khazaeli et al., 2021; Martinez-Gil, 2023).

The current model of legal services is failing to address legal needs in several important con-texts. In the United States, around 92% of civil legal problems experienced by low-income Americans receive no or inadequate legal help (Slosar, 2022). In U.S. criminal cases, where criminal defendants generally have a right to legal counsel, public defenders are systematically overworked and under-resourced (Pace et al., 2023). Access to legal services is also limited for U.S. small businesses (Baxter, 2022). The combination of unequal access to and high costs of legal services results in a troubling access to justice issue.

Internationally, there is tremendous variation in legal systems and challenges, but many opportunities where technology could help address legal inefficiencies and under-served communities have been identified globally (see e.g., Wilkins et al., 2017; Cunha et al., 2018; Bosio, 2023; World Justice Project, 2019). Against this backdrop, numerous scholars have focused on legal technology as a path towards reducing the access to justice gap by extending the abilities of legal professionals and helping non-lawyers navigate legal processes (Baxter, 2022; Bommasani et al., 2021; Cabral et al., 2012; Rhode, 2013; Katz et al., 2021).

Of course, technology is no panacea and many of the inequities and inefficiencies in jurisprudence are related to deeply rooted social and cultural phenomena. Nonetheless, we view the *responsible* deployment of legal technology as a crucial step towards improving the legal profession and broadening access to legal services.

Despite their potential to improve the provision of legal services, the legal industry has been slow to adopt computational techniques. The majority of legal services continue to rely heavily on manual work performed by highly trained human lawyers. This slow adoption may be partially attributed to risk aversion, misaligned incentives, and a lack of expertise within the legal community (Livermore and Rockmore, 2019; Fagan, 2020).

We argue that there is another factor to blame,

rooted not in legal practice but rather in legal NLP research: In short, legal NLP is failing to develop many applications that would be useful for lawyers. Instead, legal NLP research tends to focus on generic NLP tasks and applies widely-used NLP methodologies to legal data, rather than developing new NLP tools and approaches that solve problems unique to the legal context.

For example, NLP research might apply text classification to predict the direction of a U.S. Supreme Court judgment based on portions of the judicial opinion. These types of models tend to be of limited practical utility: First, the vast majority of lawyers and legal disputes will never reach the Supreme Court. Second, the legal reasoning applied by the Supreme Court is unlikely to be representative of lower courts. And lastly, classifiers trained on published judgments may emulate judicial idiosyncrasies rather than modeling optimal legal reasoning. Meanwhile, there has been less research on systems to help lawyers identify relevant precedent for trial, on exploring automated summarization and generation of legal documents, or leveraging NLP for online dispute resolution.

The work done by Livermore and Rockmore (2019); Katz et al. (2021); Cabral et al. (2012); Rhode (2013) and others takes an important step toward bridging disciplinary disconnects by providing overviews of NLP and related methods to legal and multidisciplinary communities. We hope to build on this work by encouraging the legal NLP community to understand the needs of legal practitioners. Our paper offers some initial starting points for NLP research that are informed by practical needs.

We base our argument on a review of recent legal NLP research, which identifies key themes in this literature (see Figure 1). We find that a large portion of this research focuses on tasks which we believe are disconnected from the needs of legal practitioners. We further observe that only a small fraction of citations to legal NLP publications stem from legal publications, providing evidence that NLP publications have not managed to rise to the attention of the legal community (see left panel of Figure 2). Grounded in this review, we segment legal NLP tasks into three categories: applications that could aid the provision of legal services; widespread NLP applications that have limited impact on practical legal issues; and areas of legal NLP research that could have significant impact on

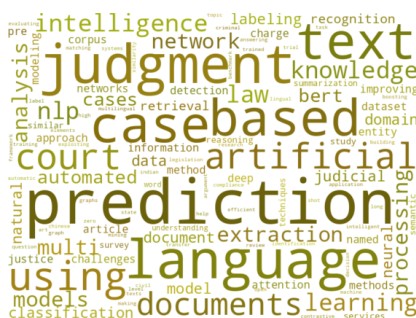

Figure 1: Word cloud of legal NLP paper titles.

legal practice but which remain underexplored.

Our work adds to a growing number of recent position papers discussing the intersection of law and NLP (see e.g., Dale, 2019; Zhong et al., 2020; Tsarapatsanis and Aletras, 2021; de Oliveira et al., 2022; Katz et al., 2023). The number of survey papers in this domain might suggest some confusion about the state of legal NLP. In addition to offering descriptive findings, we aim to provide a normative argument. In light of the access to justice issues highlighted above, we encourage legal NLP researchers to pragmatically focus on work that promises to broaden access to justice. This objective helps advance a shared normative goal that does not neglect the 'legal' dimension of legal NLP.

To summarize, we make the following contributions and recommendations.

(1) We review the current state of legal NLP.
(2) We discuss underexplored areas of legal NLP research.
(3) We propose the use of legal NLP to tackle the access to justice crisis as a shared normative goal.
(4) We advocate for more collaboration between NLP researchers and legal scholars and practitioners.

## 2 Literature Review

We conduct a rapid literature review via forward citation chasing over 171 papers, following recommendations made by Khangura et al. (2012). Literature reviews have been employed in several position papers investigating the intersection of NLP and other disciplines (see e.g., Laureate et al., 2023; Ricketts et al., 2023).

The starting point of our rapid review is Zhong

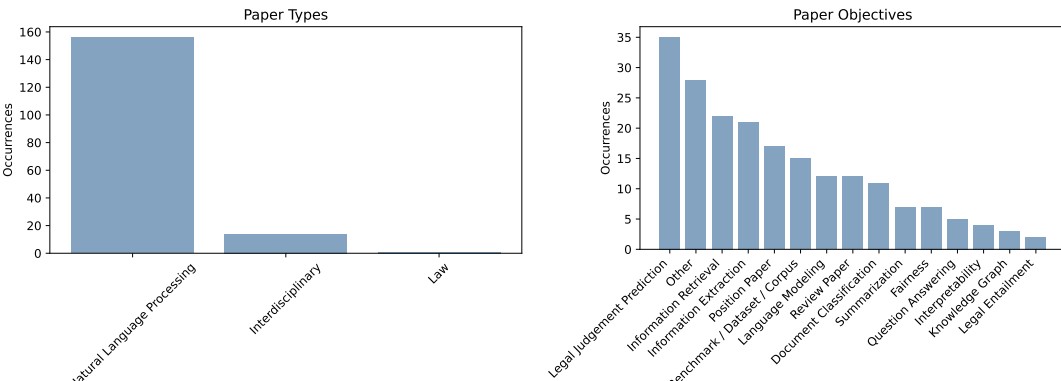

Figure 2: (Left) Bar plot of paper categories in our literature review (y-axis is the total number of papers of each type). (Right) Bar plot of objectives in the reviewed papers (y-axis is the total number of papers for each objective).

et al. (2020), an overview paper about how legal NLP benefits the legal system. We selected this work as our starting point because it is a *recent* contribution in a top NLP conference that has received a reasonably high number of citations for us to review. Moreover, it provides (to the best of our knowledge) a fairly accurate overview about the current state of legal NLP. Our review includes all papers citing Zhong et al. (2020) on Google Scholar.

Zhong et al. (2020) aim to identify popular legal NLP tasks and discuss these tasks from the perspectives of legal professionals. Tasks outlined in the paper include legal judgment prediction, question answering, and similar case matching.

We read and manually annotate all papers citing Zhong et al. (2020), resulting in 171 annotated papers.[1] For each paper, we determine:

(1) Is the paper is mainly a NLP contribution, an interdisciplinary contribution, or a legal contribution?

(2) What are the paper's key objectives (e.g., legal judgment prediction, summarization, position paper, introducing a benchmark)? A paper can have multiple objectives.

We follow a hybrid inductive-deductive content analysis methodology, employed in Birhane et al. (2022), and aim to follow best practices and recommendations from the qualitative content analysis literature (see e.g., Merriam and Grenier, 2019; Krippendorff, 2018; Hsieh and Shannon, 2005; Bengtsson, 2016).

We start with a set of ten objectives described in Zhong et al. (2020), such as relation extraction,

legal judgment prediction and question answering. We list all of these in Appendix A. If a paper describes an objective that cannot be classified into one of these categories, we define a new category. After every 20 annotations, we review all categories assigned so far and decide whether to merge existing categories (e.g., named entity recognition and semantic role labeling merge into information extraction, while similar case matching and semantic search merge into information retrieval) – this represents the deductive step in our approach. After annotating all 171 papers, we review all categories again and perform a final merge. Finally, for each paper, we determine whether it is a legal, NLP, or interdisciplinary publication. This categorization is based on the publication venue and the author's affiliations.

We display our main findings in Figure 2. We find that despite the inherently interdisciplinary nature of legal NLP, most work we reviewed is produced and consumed by the NLP community. Only 10% of the papers citing Zhong et al. (2020) are primarily interdisciplinary. Perhaps more strikingly, only a single law review article cites Zhong et al. (2020). In terms of paper objectives, we find that legal judgment prediction appears to be the most popular objective, with 20% of reviewed papers focusing on this task.

## 3 Categorizing legal NLP applications

Based on our review, we categorize popular legal NLP objectives in terms of their ability to impact the practice of law or broaden access to justice. Subsequently, we identify potentially impactful areas of research that remain underexplored.

---

[1]At the time of writing, the paper had 182 citations in total. We discarded nine citations for which Google Scholar did not provide a link.

## 3.1 Existing applications that promise to aid legal practitioners

We identify three primary streams of legal NLP research that promise to benefit legal practice and improve access to justice.

**Legal document generation and analysis.** NLP technology can help speed up drafting, e.g., via dedicated legal language models or general-purpose language models such as GPT-4. It can also help analyze legal documents, for example via information extraction or summarization (Galgani et al., 2012; Bommarito II et al., 2021). Based on a survey of senior attorneys (see Appendix B), we note that document review and generation appear to be critical tasks from the perspective of many practitioners.

**Semantic search.** Legal arguments rely heavily on citations to statutes and precedential court opinions. Several scholars have thus focused on designing systems that aid attorneys in finding citations to prior court decisions that support their arguments (Huang et al., 2021; Tang and Clematide, 2021).

**Accessibility of legal language.** The translation of legal jargon into more accessible forms has been identified as an important priority by legal scholars (Benson, 1984). Here, style transfer methods, legal summarization, question answering and information extraction methods can all prove helpful to make legal language more accessible and to surface key concepts (see e.g., Farzindar and Lapalme, 2004; Manor and Li, 2019; Khazaeli et al., 2021). These applications can help judges quickly understand filings submitted by attorneys, aid lawyers in gaining an overview of documents, and can help individuals better understand contracts, wills and other documents that may be relevant to them.

## 3.2 Applications that fail to aid legal practitioners

Some legal NLP publications focus on tasks that are simply not part of legal practice or that use legal data in ways that do not fully account for how this data was generated. Other publications focus on tasks with significant ethical implications that make them ill-suited for real-world deployment.

Legal judgment prediction (LJP), the most common task identified in our review, suffers from both of these weaknesses. First, LJP typically extracts facts from court opinions and then uses the facts to predict the associated judgment. This approach is problematic because the narrative presented by judges in their opinions is typically crafted with an outcome in mind, thereby precluding neutrality in the facts they present. As such, LJP treats human-generated annotations as ground truths when in fact these annotations are based on confounding factors. Moreover, the automation of legal judgments is fraught with ethical challenges. Biased judgments would have grave social implications, not only for litigants directly affected by inaccurate legal judgments but also society at large if automated judgments undermine trust in the judicial system. LJP may have utility for low-stakes disputes that may not otherwise see a day in court, or it could be used to simulate a specific judge's or court's idiosyncrasies, which may be a helpful strategic tool for potential litigants. Furthermore, LJP might also be useful to surface existing biases in judicial decision making. However, LJP is typically framed as modeling the "correct" application of laws to facts. Due to its inherent risks, this application should be carefully considered and it is unlikely to materialize in the near future, if at all.

It is important to underscore that other common legal NLP tasks may not directly aid legal practitioners, but nevertheless provide valuable resources and insights. These include detecting bias in legal language (Rice et al., 2019) and legal NLP benchmarks which help measure the progress of NLP methods (Chalkidis et al., 2022; Guha et al., 2023).

## 3.3 Underexplored applications that promise to aid legal practitioners

Understanding the nature of legal practice in more detail can help surface applications of NLP that would be useful to legal practitioners. Of course, this is easier said than done as there are still limited opportunities for legal NLP researchers and legal practitioners to exchange ideas. For this discussion, we draw partially on a survey conducted as part of Harvard Law School's 2023 *Leadership in Law Firms* program (LLF). This survey asked over 50 senior attorneys from 17 different countries to identify potentially impactful applications of NLP which would provide value in their firms (see Appendix B for an overview of responses).[2]

---

[2] We recognize that these responses are not representative of legal practice generally, but present them as a valuable example of how practitioners think about NLP and as a starting point for ideation.

**Persuasive legal reasoning.** Litigation is at least partially a rhetorical exercise in which attorneys seek to identify the most persuasive arguments while taking into account the presiding judge and, in some instances, the composition of the jury. The nature of litigation offers ample opportunity for the study of language in general, and the study of discourse and pragmatics specifically. Extraneous factors, like the presiding judge, have a significant impact on the persuasiveness of different arguments, and there already exists NLP research on context-aware argumentation (see e.g. Durmus et al., 2019) that could be applied to law.

**Practice-oriented legal research tools.** Legal research and case analysis was one of the key areas identified in the LLF Survey. In common law jurisdictions, law develops organically through judicial opinions and higher courts may overturn or refine past court decisions. Legal research platforms label whether a case is considered "good law", that is whether it remains a good basis for future arguments and thus current law. Current citation prediction work has largely ignored this aspect, creating a risk that outdated or overturned opinions are recommended. NLP research techniques such as sentiment analysis could identify *how* a citation is used by judges to determine whether it remains good law.

A related extension is the retrieval of individual legal passages. Judicial opinions are generally long documents and legal practitioners normally cite very specific passages. As a result, legal research platforms often present specific passages as "head notes" or "key cites" to allow lawyers and judges to identify the most important portions of opinions. Legal passage prediction (LPP) seeks to predict specific passages, rather than entire judicial opinions, which is more closely aligned with the needs of legal professionals (Mahari, 2021). LPP may also be combined with extractive summarization (see e.g. Bauer et al., 2023), to identify passages from an opinion that are most likely to represent useful citations.

**Retrieval augmented generation over private legal data.** A more general opportunity for legal NLP is related to proprietary legal data. Law firms amass large knowledge banks from past cases that contain sensitive and confidential data. Practicing attorneys routinely build on their past work and experience. NLP tools could help them identify relevant records and, based on these retrieved records, generate new documents. Retrieval augmented generation (see e.g. Lewis et al., 2020; Borgeaud et al., 2022; Shi et al., 2023) is well suited to this task, however, it is critical that confidential records are not leaked to external parties or other public databases (Arora et al., 2023), and that generation is performed in an auditable fashion (Mahari et al., 2023).

## 4 Discussion

The disconnect between AI research on applications and specific disciplines is not limited to law (see e.g. Acosta et al., 2022). Law, however, is unique among disciplines in that it is a field built on language. Given the current state of legal practice, there is a need for innovation to make legal services more affordable and to address the access to justice crisis. As such, law presents a valuable opportunity for the NLP community to conduct research on applications that could aid legal practitioners and that expand access to justice.

Impactful legal NLP research must be grounded in the needs of the legal community. The observed lack of cross-disciplinary citations in our review suggests that legal NLP researchers are largely disconnected from the legal community. We encourage legal NLP researchers to identify tasks that are performed frequently by legal practitioners and that lend themselves to the application of NLP techniques. To aid NLP researchers in identifying these tasks, we urge them to consider closer interdisciplinary collaborations with the legal community or at least to address legal issues identified in the legal literature.

## 5 Conclusion

By leveraging a literature review, we find that the legal NLP community is largely disconnected from legal academia. We emphasize that certain popular legal NLP tasks are only of limited utility to legal practitioners. We thus urge legal NLP researchers to focus on access to justice as a shared normative objective, to ground their work in the needs of the legal community, and to consider collaborating with lawyers to ensure that their research has applications in practice. NLP has the potential to positively transform the practice of law and by extension society. However, this is impossible without cross-disciplinary understanding and collaboration.

## Limitations

**Business applications.** Reviews of NLP literature provide insight into academic work, but they do not reveal business applications of NLP. While we observe a disconnect between NLP research and law in academia, it is possible that there exists unpublished work that is more attuned to the needs of the legal community. However, this work tends to focus on profitable applications, which are not always aligned with broadening access to justice. The LLF survey provides a business-oriented perspective by surveying senior attorneys from an international group of law firms, however, more exhaustive work is needed to fully understand where NLP tools provide value to law firms and to what degree these offerings also address access to justice issues.

**Scope.** We conduct a rapid review based on citations to a popular NLP paper. Our intuitions about the field lead us to believe that our findings extrapolate to the field as a whole. Contemporaneous work provides a broader overview and identifies similar trends as our review. For example, Katz et al. (2023) find that classification is the most popular objective in the legal NLP community (LJP represents a classification task). While our work is similar in spirit to Katz et al. (2023), we take a less descriptive but more normative approach.

## Ethics Statement

If aligned with societal needs, legal NLP has tremendous potential to expand access to justice, to reduce biases in legal practice, and to create new efficiencies in legal practice. At the same time, legal NLP deals with a sensitive aspect of society. Poorly designed NLP tools could embed biases, remove human oversight, or undermine trust in the legal system. Our hope is that encouraging collaboration between NLP and legal researchers will also help identify and mitigate ethical challenges.

**Broader Impact.** This publication is a perspective about the current state of legal NLP and its future directions, grounded in evidence about interdisciplinary disconnects. Of course, the trajectory of an academic field ought to be based on deliberative discussions involving many stakeholders. We present our recommendations and visions about the future of legal NLP, which are, at least to some extent, subjective. We invite others to expand on and to critique our views and hope to contribute to a broad and thoughtful discussion about the future of legal NLP.

## Acknowledgements

We are grateful to the Harvard Law School Center on the Legal Profession, Scott Westfahl, and David Wilkins for helping us gain insight into the needs of legal practitioners; we appreciate all participants in the Law Firms Leadership Program for taking time to share their perspectives with us. We thank Ron Dolin for his extremely helpful feedback on this paper.

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

## A  Initial Objectives

Relation Extraction
Event Timeline
Element Detection
Legal Judgment Prediction
Question Answering
Similar Case Matching
Summarization
Embeddings
Knowledge Graphs
Language Models

Table 1: Set of initial objectives for the literature review.

## B  Harvard Leadership in Law Firms: Survey Responses

As part of Harvard Law School's *Leadership in Law Firms* program, the lead author of this paper conducted a survey of over 50 senior attorneys from 17 different countries on applications of NLP in legal practice. The survey asked: *What is one legal-related task (e.g., document review, responding to a motion), system (e.g., time entry or giving feedback), type of legal matter (deals, regulatory review) that you would LOVE for generative AI to make easier/more efficient?*

While by no means representative of the legal industry as a whole, the survey responses provide valuable insight into the priorities of practicing attorneys. As such, they serve as a starting point for some new avenues of legal NLP research and an example of how the research community can solicit insights from practitioners.

At a high level, the following application categories emerged from the survey:

(1) Legal and business document review and summarization (42/59)
(2) Time entry and billing, case intake, and reviewing invoices (14/59)
(3) Case law research and regulatory review (11/59)
(4) Legal document generation, creating multi-document summaries (3/59)

(5) Simulating or predicting legal outcomes (2/59)
(6) Project and process management (2/59)
(7) Knowledge management (1/59)

**Raw Responses**
1. Document review and summaries
2. Case review intake
3. Initial research from multiple sources to create first draft memo. For financial services regulatory
4. Document review, precedents, process improvement
5. Time entry
6. Case analysis and statistics; usage in the discovery process
7. Billing Process
8. Time entry
9. Summarise an extensive document or prepare a well substantiated research
10. Financial/tax modelling, document review
11. Documentation review, regulatory review
12. Analysis of high volume procedural evidence
13. Document review
14. Time entry
15. Legal research (e.g., finding relevant case law)
16. Time recording
17. Predictive tool for client outcomes
18. Document review
19. Time entry
20. Documents review
21. Document and regulatory review
22. Document review in large, complex litigation
23. First draft of letters to media and social media
24. Review of trading data in securities enforcement matters
25. Time entry and billing
26. Legal research and regulatory option
27. Use of AI in the analysis of the firm's own data sets in order to make use of expertise available in the firm as quickly and effectively as possible, such as previously prepared expert opinions on a topic. Furthermore, it should be possible to search external databases as effectively as possible
28. Document review
29. Document review
30. Dealing with AML and KYC obligations
31. Document review
32. Time recording
33. Admin related tasks around time entry, review of accounts
34. One reliable source with brief and regular updates on case-law, legislation and important developments including access to in-depth information for the individual sources of law
35. Brief writing, time entry, legal research
36. Document Review and Legal Search - time entry - regulatory review
37. Lease summaries
38. Document review
39. Time entry
40. Giving feedback
41. Automating tasks/workflows in the sense of having a spread sheet/document assistant
42. Document review in diligence processes
43. Time keeping
44. Document Review
45. Document review (both consulting and litigation)
46. Document review
47. Document review
48. A system providing full and reliable overviews on legal topics by analyzing all relevant sources including legislation, legislation processes, case law and literature. This would help to often spend long time on getting certainty about being up to date
49. Document review
50. Comparative summary of publications and judgments

51. Horizon scanning for regulatory change
52. Document review
53. Analysis and comparison of different information sources, Intranet, Internet, databases
54. Document review in both counseling and litigation
55. Document review, regulatory review
56. Document Review
57. Legal-related task: document review and research; System: project management; Legal Matter: Due Diligence/deals
58. Administrative things like time entry or reviewing invoices
59. Summarise big volume of data