# OpenReview forum: "The Law and NLP: Bridging Disciplinary Disconnects"
_EMNLP/2023/Conference — EMNLP 2023 Findings_

### Official Review · Reviewer_aLn7 · 2023-07-20

**Soundness:** 2

**Excitement:**

3: Ambivalent: It has merits (e.g., it reports state-of-the-art results, the idea is nice), but there are key weaknesses (e.g., it describes incremental work), and it can significantly benefit from another round of revision. However, I won't object to accepting it if my co-reviewers champion it.

**Paper Topic And Main Contributions:**

The survey paper analyses past legal NLP papers and their topic, and identifies which tasks are the most relevant to legal practitioners. The problem that the authors identify is that there is often a gap between legal nlp tasks and concerns and the requirements of legal work.

**Questions For The Authors:**


- What is "interdisplinary" (line 159)? Can you specify what is meant here: e.g. the publication venue is not computer science/nlp, or authors of a paper are affilicated with different departements?

- Figure 2, right side: what is in "other"? it is the second category but we don't have precisions on what is included  in there.

- The paragraph starting line 285 suggests that NLP researcher create more opportunity to acces proprietary data. Can you specify?


**Reasons To Accept:**

- The purpose of the paper is very relevant and has the potential to trigger and frame future research.

- Legal NLP to expose biases is very useful (line 235), as legal decisions and algorithmic legal decisions have been shown in the past to hold and perpetuate biases against marginalised communities and already vulnerable groups. It is pointed out by the authors that LPJ  is not the most useful task and there is good arguments. However it should be said in nuance that LPJ aim is not always to automate judgements, but to expose biases for example (Early Predictability of Asylum Court Decision, Dunn et al.)

- The authors point out that 2 tasks that are under-studied yet the most useful for the legal work citations extraction, and legal passage prediction. There are good arguments as to why those 2 tasks are essential (line 276) from the legal practitioners. Similarly, access to justice is under-studied in legal NLP, although it may be one domain where NLP could really make a difference thanks to scalability.



**Reasons To Reject:**

- The paper analysed literature that is collected if citing one specific paper (Zhong et al.) which sounds restrictive as mentioned in the limitations section by the authors themselves.

- There are other restrictions to the application of NLP in law that are hardly mentioned, notably the access to clean data and the access to annotated data. There are few legal datasets, and especially there are few legal datasets that are curated for research work. There is a large amount of publicly accessible data published online that is not easy to process for research (OCR documents for example).
This is mentioned in the paragraph starting line 285 through the prism of proprietary data access. However, regarding this paragraph: there is legal NLP research in industry as well that is regularly published in NLP outlets. While the paragraph makes a valid point, the use of certain legal data causes commercial issues, IP issues, and potentially privacy issues that also need to be taken into account.

- There is strong point being made on LPJ being irrelevant to legal work. While this may sometimes be true, it is interesting to distinguish areas of law: it may be useful for a company to know if it is worth filing a case or not, or to know the amount of work that you need to put in depending the chances of success. LPJ may not have been done on datasets that seem aligned with this goal (eg SCOTUS), but 1/ this is taking us back to few datasets available and 2/ once the methodology has been explore on one dataset, one could argue that is is easier to replicate. Therefore it could be interesting to nuance this position.

- Annotations of eg. events, names, locations are ground truth. The authors make the point that despite being correctly annotated, in may be the case that facts presented are incomplete and partial. While this might be true in some cases, it does not mean necessarily that the annotations are incorrect and don't have ground truth. This also assume that the LPJ is based on one single report document, when it may also process several documents (multi-document) that constitute the file. Therefore this argument seems weak.
On the problem of the truth of the annotations, it would be worth mentioning that when there is in fact no ground truth (annotation the reasons for the judgment, a legal argument for instance) the lack of clarity results more often from human annotator disagreeing on the qualification of an argument/token.

Overall the paper has a very interesting starting point and purpose that would hugely benefit the legal NLP community. However it lacks clarity, and would benefit from being more grounded in the legal nlp literature and should aim to be exhaustive both on the analysed literature and in support to the normative arguments of the authors.


**Reproducibility:**

N/A: Doesn't apply, since the paper does not include empirical results.

**Reviewer Confidence:**

5: Positive that my evaluation is correct. I read the paper very carefully and I am very familiar with related work.

**Typos Grammar Style And Presentation Improvements:**

"The Law and NLP – Bridging Disciplinary Disconnects" is the title on the PDF, the title on OpenReview is: "Legal Passage Retrieval: A pragmatic approach to legal AI"

---

> ### Author Rebuttal · Authors · 2023-08-28
>
> Many thanks for the exhaustive and constructive review which is highly appreciated!
> - To clarify our approach: We have taken the Zhong et al. 2020 paper as a starting point and then conducted a rapid literature review of the papers that cite the Zhong et al. paper, which, by construction, were all written in 2020 or later. From the 170 papers examined, roughly half are quite recent – from 2022 or 2023. Further, this set of papers includes (in our judgment) the most relevant new research in legal NLP, including (but not limited to) “LexGLUE” (Chalkidis et al. 2021), “GPT-4 passes the bar exam” (Katz et al 2023), and “LegalBench” (Guha et al. 2023). Hence, we feel that this set of papers fairly represents the legal NLP field. A more recent broad legal NLP review, “Natural Language Processing in the Legal Domain” (Katz et al. 2023), has only been cited seven times and those papers also cite the Zhong et al. paper. While it would be possible to further extend or restrict this list in various ways, this would be ad-hoc, subjective, and unlikely to be more representative of new research in the field. Our approach is based on best practices for conducting such systematic literature reviews.
> - Lack of discussion of other barriers to applying NLP in law: Thank you for this comment! We expand on these restrictions in the final version of the paper, and also highlight some of the industry publications for a more fine-grained discussion, e.g., Norkute et al., 2021 "Towards Explainable AI: Assessing the Usefulness and Impact of Added Explainability Features in Legal Document Summarization"
> - Relevance of LJP to legal practice: Thank you for your comment, other reviewers have provided similar feedback. In our revised manuscript we include a more nuanced discussion about the relevance of LJP to legal practice, in particular, the potential of LJP to be used to surface biases in judicial decision-making. We underscore that there are merits to LJP and that future research should expand on these. However, we do not view LJP as “the holy grail” of legal NLP and feel that common motivations for LJP fail to address its real-world implications and limitations, for example, "Legal judgment prediction (LJP) is a fundamental task in legal AI, which aims to assist the judge to hear the case and determine the judgment." (Wu et al. 2022). We are primarily advocating for more interaction between legal practitioners and legal NLP researchers, which, among other things, involves a discussion about the merits and limitations of LJP, especially within the legal NLP community.
> - Regarding ground-truth annotations: While ground-truth labels exist, they are not always straightforward. The underlying documents, e.g., facts in a judicial opinion, are presented in ways that support the ultimate judgment. If the judgment were different, the presentation of facts likely would be different too. In practice, judgments are subjective, and it is often unclear whether two judges would come to the same conclusions, making the “ground truth” more complex than it first appears. Instead of thinking in terms of ground truths, it can be helpful to think about adversarial perspectives, for example by finding agreement on facts between the two opposing legal briefs. As you point out, carefully curated multi-document approaches can be helpful, although we find that even when LJP approaches are based on multiple documents, they commonly fail to take full account of how the documents were generated and what biases they may inherently contain. We clarify these points in the final version and again advocate for more communication between legal practitioners and legal NLP researchers to avoid such sources of confusion in the future.
>
> Answers to your questions:
> - “Interdisciplinary” is based on publication outlets and/or author affiliations. A paper is labeled as interdisciplinary either if it appears in an interdisciplinary venue or if it has authors from both law schools and computer science departments.
> - The “other” category consists of 27 singleton items, e.g., part of speech tagging in the legal domain. We include all of these in the Appendix in the final version of the paper.
> - Regarding opportunities to access proprietary legal data: As you also highlight in your review, one of the key challenges for legal NLP is access to (clean) data. In working with legal practitioners, we have found that researchers can gain access to high-quality proprietary data that enables fruitful legal NLP work. This poses a challenge to the community: While there are excellent justifications for norms against publications that rely on proprietary data, the legal NLP community may benefit from providing at least some opportunities to publish contributions that would be infeasible without the use of such data. More generally, our revised manuscript contains an expanded discussion of data availability as a major challenge for legal NLP, building on work done in this area such as “How to build a more open justice system” (Pah et al. 2020).

---

### Official Review · Reviewer_V1t3 · 2023-07-28

**Typos Grammar Style And Presentation Improvements:** The style is up to standard and, abov…
**Soundness:** 5

**Excitement:**

4: Strong: This paper deepens the understanding of some phenomenon or lowers the barriers to an existing research direction.

**Missing References:**

There are no references that I can think of which are missing.

**Paper Topic And Main Contributions:**

This is a position paper that very helpfully sets out ways in which 'legal NLP' could develop so as to help legal practitioners (and perhaps academics), in areas other than predictive tasks. The paper's main contribution is that it criticises very clearly extant research trends and also very clearly says which areas of future research would be most helpful for practitioners.

**Questions For The Authors:**

Question 1A: Do you think that there is some way in which predictive tasks could be somehow salvaged? I am thinking here of either prediction that uses metadata of judgments (e.g. judges' names) or other, potentially helpful, kinds of textual data.

**Reasons To Accept:**

The paper makes a significant contribution to oingoing debates, because (a) it summarises existing literature very well and (b) makes a clear (and, to this reviewer's mind, persuasive) case for a re-orientation of legal NLP. It achieves the above in a clear and concise way. It should therefore be accepted.

**Reasons To Reject:**

I see no reasons to reject the paper. On the contrary, I think that it deserves a wide discussion.

**Reproducibility:**

N/A: Doesn't apply, since the paper does not include empirical results.

**Reviewer Confidence:**

5: Positive that my evaluation is correct. I read the paper very carefully and I am very familiar with related work.

---

> ### Author Rebuttal · Authors · 2023-08-28
>
> Thank you for the very positive review! To address your question somewhat qualitatively: we believe that predictive tasks should for now mostly be used to investigate existing biases in the legal system, rather than as autonomous decision-making systems, which could have far-reaching negative externalities if deployed (see e.g., the ProPublica investigation of the COMPAS tool). Specifically, prediction tools that use features that ought to be irrelevant to a given decision (e.g., past decisions or judges’ public remarks) can help expose biases. To this extent, metadata on judgments can and should certainly be used, e.g., as fixed effects. That said, “Human Decisions and Machine Predictions" (J Kleinberg et al. 2017) suggests that prediction could help judges render “correct” decisions, and given adequate safeguards it is possible that automated judgment prediction will one day augment judges.
>
> We incorporate these recommendations in the revised version of the paper.

---

### Official Review · Reviewer_6D6N · 2023-08-04

**Soundness:** 3

**Excitement:**

3: Ambivalent: It has merits (e.g., it reports state-of-the-art results, the idea is nice), but there are key weaknesses (e.g., it describes incremental work), and it can significantly benefit from another round of revision. However, I won't object to accepting it if my co-reviewers champion it.

**Paper Topic And Main Contributions:**

This paper makes the case that legal practising still involves a significant amount of human labour, in part because popular NLP tools being tailored to legal documents rather than new NLP tools being created to address the specific challenges faced by the legal profession.

**Reasons To Accept:**

The claims made are valid in many domains; however, in this case, they are based on a systematic review of over 170 scientific articles.

**Reasons To Reject:**

The study is based on the articles reviewed in Haoxi Zhong, Chaojun Xiao, Cunchao Tu, Tianyang 509 Zhang, Zhiyuan Liu, and Maosong Sun. 2020. In the past three years, however, NLP and AI have developed very quickly, and there is certainly a lot of new research in the field. This raises questions about the conclusions reached.

**Reproducibility:**

N/A: Doesn't apply, since the paper does not include empirical results.

**Reviewer Confidence:**

1: Not my area, or paper was hard for me to understand. My evaluation is just an educated guess.

---

> ### Author Rebuttal · Authors · 2023-08-28
>
> Many thanks for this constructive comment. To clarify our approach, we have taken the Zhong et al. 2020 paper as a starting point and then conducted a rapid literature review of the papers that cite the Zhong et al. paper, which, by construction, were all written in 2020 or later. From the 170 papers examined, roughly half are quite recent – from 2022 or 2023. Further, this set of papers includes (in our judgment) the most relevant new research in legal NLP, including (but not limited to) “LexGLUE” (Chalkidis et al. 2021), “GPT-4 passes the bar exam” (Katz et al 2023), and “LegalBench” (Guha et al. 2023). Hence, we feel that this set of papers fairly represents the legal NLP field. A more recent broad legal NLP review, “Natural Language Processing in the Legal Domain” (Katz et al. 2023), has only been cited seven times and those papers also cite the Zhong et al. paper. While it would be possible to further extend or restrict this list in various ways, this would be ad-hoc, subjective, and unlikely to be more representative of new research in the field. Our approach is based on best practices for conducting such systematic literature reviews.
>
> We make these points more explicit in a revised version and hope that this remark somewhat alleviates your concerns about the validity of the conclusions, especially with respect to the more recent developments in NLP and AI for law.

---

### Meta-Review · Area_Chair_XaW4 · 2023-09-14

**Recommendation:** 3

**Metareview:**

This is a short paper containing a blend of a (somewhat) systematic literature review and a position paper.
There is quite a gap between the scores of the reviewers that is mostly due to how to value these contributions, while they seem to agree about the "facts".

In my view, it would have been better to either write a more systematic literature review or a real position paper.

---

### Decision · Program_Chairs · 2023-10-07

**Decision:**

Accept-Findings

**Comment:**

This is a short paper containing a blend of a (somewhat) systematic literature review and a position paper.
There is quite a gap between the scores of the reviewers that is mostly due to how to value these contributions, while they seem to agree about the "facts".

In my view, it would have been better to either write a more systematic literature review or a real position paper.